# DiFSD: Ego-Centric Fully Sparse Paradigm with Uncertainty Denoising and Iterative Refinement for Efficient Self-Driving

## Abstract

Current end-to-end autonomous driving methods resort to unifying modular designs for various tasks (e.g. perception, prediction and planning). Although optimized in a planning-oriented spirit with a fully differentiable framework, existing end-to-end driving systems without ego-centric designs still suffer from unsatisfactory performance and inferior efficiency, owing to the rasterized scene representation learning and redundant information transmission. In this paper, we revisit the human driving behavior and propose an ego-centric fully sparse paradigm, named DiFSD, for end-to-end self-driving. Specifically, DiFSD mainly consists of sparse perception, hierarchical interaction and iterative motion planner. The sparse perception module performs detection, tracking and online mapping based on sparse representation of the driving scene. The hierarchical interaction module aims to select the Closest In-Path Vehicle / Stationary (CIPV / CIPS) from coarse to fine, benefiting from an additional geometric prior. As for the iterative motion planner, both selected interactive agents and ego-vehicle are considered for joint motion prediction, where the output multi-modal ego-trajectories are optimized in an iterative fashion. Besides, both position-level motion diffusion and trajectory-level planning denoising are introduced for uncertainty modeling, thus facilitating the training stability and convergence of the whole framework. Extensive experiments conducted on nuScenes dataset demonstrate the superior planning performance and great efficiency of DiFSD, which significantly reduces the average L2 error by **66%** and collision rate by **77%** than UniAD while achieves **8.2×** faster running efficiency.

## 1 Introduction

Autonomous driving has experienced notable progress in recent years. Traditional driving systems are commonly decoupled into several standalone tasks, e.g. perception, prediction and planning. However, heavily relying on hand-crafted post-processing, the well-established modular systems suffer from information loss and error accumulation across sequential modules. Recently, end-to-end paradigm integrates all tasks into a unified model for planning-oriented optimization, showcasing great potential in pushing the limit of autonomous driving performance.

Literally, existing end-to-end models Hu et al. (2023); Ye et al. (2023); Jiang et al. (2023); Sun et al. (2024) designed for reliable trajectory planning can be classified into two mainstreams as summarized in Fig. 1(a) and (b). The dense BEV-Centric paradigm Hu et al. (2023); Ye et al. (2023) performs perception, prediction and planning consecutively upon the shared BEV (Bird's Eye View) features, which are computationally expensive leading to inferior efficiency. The sparse Query-Centric paradigm Sun et al. (2024) utilizes sparse representation to achieve scene understanding and joint motion planning, thus improving the overall efficiency. However, object-intensive motion prediction inevitably causes computational redundancy and violates the driving habits of human drivers, who usually only concentrate on the Closest In-Path Vehicle / Stationary (CIPV / CIPS) which are more likely to affect the driving intention and trajectory planning of ego-vehicle. Meanwhile, excessive interaction with irrelevant agents will be conversely adverse to the ego-planning. Therefore, the planning performance remains unsatisfactory in both planning safety, comfort and personification.

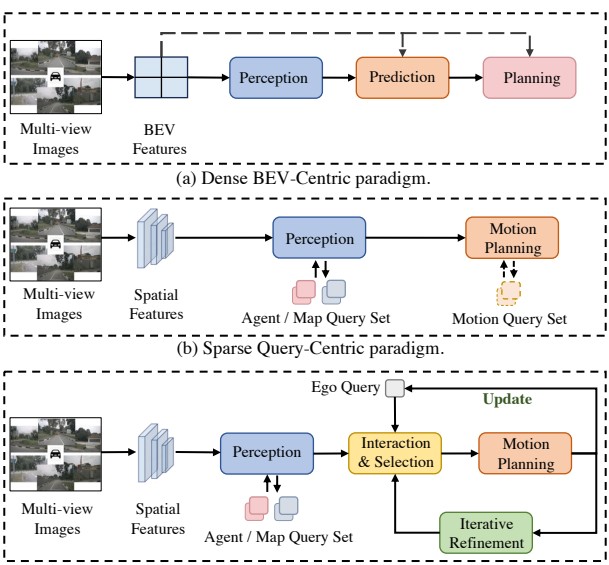

Figure 1: The comparison of different end-to-end paradigms. (a) The dense **BEV-Centric** paradigm. (b) The sparse **Query-Centric** paradigm. (c) The proposed fully sparse **Ego-Centric** paradigm.

To this end, we propose DiFSD, an Ego-Centric fully sparse paradigm as shown in Fig. 1(c). Specifically, DiFSD mainly consists of sparse perception, hierarchical interaction and iterative motion planner. In the sparse perception module, multi-scale image features extracted from visual encoder are adopted for object detection, tracking and online mapping simultaneously in a sparse manner. Then the hierarchical interaction performs ego-centric and object-centric dual interaction to select the CIPV / CIPS with the help of an additional geometric prior. Thus the interactive queries can be selected gradually from coarse to fine. As for the motion planner, the mutual information between sparse interactive queries and ego-query is considered for motion prediction in a joint decoder, which is neglected in previous methods Hu et al. (2023); Jiang et al. (2023) but is essential especially in the scenarios like intersections. To ensure the planning rationality and selection accuracy of interactive queries, the iterative planning optimization is further applied to the multi-modal proposal ego-trajectories, through continually updating the reference line and ego-query. Moreover, both position-level motion diffusion and trajectory-level planning denoising are introduced for stable training and fast convergence. It can not only model the uncertain positions of interactive agents for motion prediction, but also enhance the quality of trajectory refinement with arbitrary offsets. With above elaborate designs, DiFSD exhibits the great potential of fully sparse paradigm for end-to-end autonomous driving, which significantly reduces the average L2 error by **66%** and collision rate by **77%** than UniAD Hu et al. (2023) respectively. Notably, our DiFSD-S achieves **8.2×** faster running efficiency as well. In sum, the main contributions of our work are as follows:

- We propose an ego-centric **Fully Sparse** paradigm for end-to-end self-**D**riving, named as **DiFSD**, without any computationally intensive dense scene representation learning and redundant environmental modeling, which is proven to be effective and efficient for path planning of ego-vehicle.

- We introduce a geometric prior through intention-guided attention, where the **Closest In-Path Vehicle / Stationary (CIPV / CIPS)** are gradually picked out through ego-centric cross attention and selection. Besides, both position-level diffusion of interactive agents and trajectory-level denoising of ego-vehicle are adopted for uncertainty modeling of motion planning respectively.

- Extensive experiments are conducted on the famous nuScenes Caesar et al. (2020) dataset for planning performance evaluation, which demonstrate the superiority and prominent efficiency of our DiFSD method, revealing the great potential of the proposed ego-centric fully sparse paradigm.

## 2 RELATED WORK

### 2.1 END-TO-END PERCEPTION

Recent years witness remarkable progress achieved in multi-view 3D detection, which mainly build elaborate designs upon the dense BEV (Bird's Eye View) or sparse query features. To generate BEV features, LSS Philion & Fidler (2020) lifts 2D image features to 3D space using depth estimation

results, which are then splatted into BEV plane. Follow-up works apply such operation to perform view transform for 3D detection task, and have made significant improvement in both detection performance Huang et al. (2021); Huang & Huang (2022a); Li et al. (2023); Han et al. (2024) and efficiency Liu et al. (2023d); Huang & Huang (2022b). Differently, some works Li et al. (2022b); Yang et al. (2023); Huang et al. (2023) project a series of predefined BEV queries in 3D space to the image domain for feature sampling. As for the sparse fashion, current methods Wang et al. (2022); Liu et al. (2022; 2023c;a); Lin et al. (2023) adopt a set of sparse queries to integrate spatial-temporal aggregations from multi-view image feature sequence for iterative anchor refinement, where the advanced queries adopted in Liu et al. (2023a); Lin et al. (2023) contain both explicit geometric anchors and implicit semantic features.

Besides, Multi-Object Tracking (MOT) across multi-cameras is also required for downstream tasks. Traditional algorithms Wang et al. (2023); Yin et al. (2021); Weng et al. (2020) resort to "tracking-by-detection" paradigm, which relies on hand-crafted data association between the tracked trajectories and new-coming perceived objects. Recent works Zeng et al. (2022); Zhang et al. (2023); Yu et al. (2023); Meinhardt et al. (2022); Sun et al. (2012) seek to explore the joint detection and tracking methods by introducing track queries to detect the unique instances continuously and consistently. Sparse4Dv3 Lin et al. (2023) proposes an advanced 3D detector which takes full advantage of spatial-temporal information to propagate temporal instances for identity reserving, thus achieving superior end-to-end performance without additional tracking designs.

## 2.2 ONLINE MAPPING

Maps could provide important static scenario information to ensure driving safety. Current works Li et al. (2022a); Liu et al. (2023b); Liao et al. (2022); Yuan et al. (2024) manage to construct online maps with on-board sensors, instead of using HD-Map which is labor intensive and expensive. HDMapNet Li et al. (2022a) achieves this aim through BEV semantic segmentation and heuristic post-processing to generate map instances. VectorMapNet Liu et al. (2023b) introduces a two-stage auto-regressive transformer to refine map elements consecutively. MapTR Liao et al. (2022) regards map elements as a set of points with equivalent permutations, while StreamMapNet Yuan et al. (2024) adopts a temporal fusion strategy to enhance the performance. However, all of them reply on dense BEV features for online map construction, which is computationally intensive and not extensible to the sparse manner.

## 2.3 END-TO-END MOTION PREDICTION

Motion prediction of surrounding agents in an end-to-end fashion can relieve the accumulative error between standalone models. FaF Luo et al. (2018) predicts both current and future bounding boxes from images using a single convolution network. IntentNet Casas et al. (2018) attempts to reason high-level behavior and long-term trajectories simultaneously. PnPNet Liang et al. (2020) aggregate trajectory-level features for motion prediction through an online tracking module. ViP3D Gu et al. (2023) takes images and HD-Map as input, and adopts agent queries to conduct tracking and prediction. PIP Jiang et al. (2022) further proposes to replace HD-Map with local vectorized map.

## 2.4 END-TO-END PLANNING

End-to-end planning paradigm either unites modules of perception and prediction Hu et al. (2023); Jiang et al. (2023); Zhang et al. (2024); Ye et al. (2023), or adopts a direct optimization on planning without intermediate tasks Codevilla et al. (2018; 2019); Prakash et al. (2021), which lack interpretability and are hard to optimize. Recently, UniAD Hu et al. (2023) presents a planning-oriented model which integrates various tasks in the dense BEV-Centric paradigm, achieving convincing performance. VAD Jiang et al. (2023) learns vectorized scene representations and improves planning safety with explicit constraints. GraphAD Zhang et al. (2024) constructs the interaction scene graph to model both dynamic and static relations. SparseDrive Sun et al. (2024) introduces the sparse perception module for parallel motion planner. However, using straightforward designs and exhaustive modeling without ego-centric interaction, will inevitably lead to unsatisfactory planning performance and inferior efficiency.

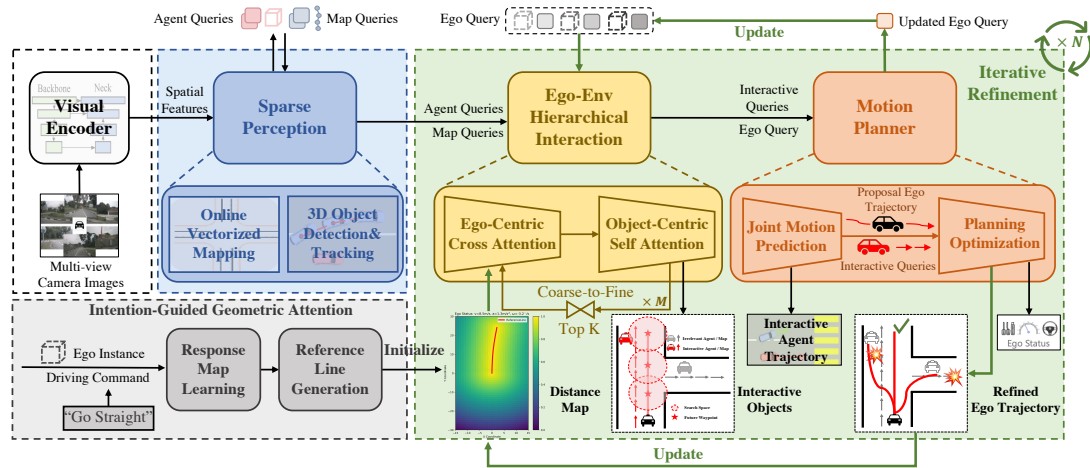

Figure 2: Overview of our proposed framework. DiFSD first extracts multi-scale image features from multi-view images using an off-the-shelf visual encoder, then perceives both dynamic and static elements in a sparse manner. The Ego-Env hierarchical interaction module is presented to select the interactive queries from coarse to fine using three different driving commands of ego queries, which are leveraged for joint motion planner through iterative refinement. An additional geometric prior is introduced for high-quality query ranking through intention-guided attention. Besides, both position-level agent diffusion and trajectory-level ego-vehicle denoising are conducted for uncertainty modeling of the end-to-end driving system.

# 3 OUR APPROACH

## 3.1 OVERVIEW

The overall framework of DiFSD is illustrated in Fig. 2, which deals with the end-to-end planning task in an ego-centric fully sparse paradigm. Specifically, DiFSD mainly consists of four parts: visual encoder, sparse perception, hierarchical interaction and iterative motion planner. First, the visual encoder extracts multi-scale spatial features from given multi-view images. Then the sparse perception takes the encoded features as input to perform detection, tracking and online mapping simultaneously. In the hierarchical interaction module, the ego query equipped with a geometric prior is introduced to pick out the interactive queries through ego-centric cross attention and hierarchical selection. In the iterative motion planner, both interactive agents and ego-vehicle are considered for joint motion prediction, then the predicted multi-modal ego-trajectories are further optimized iteratively. Meanwhile, both position-level diffusion of interactive agents and trajectory-level denoising of ego-vehicle are conducted for uncertainty modeling of motion and planning tasks respectively.

## 3.2 PROBLEM FORMULATION

Given multi-view camera image sequence can be denoted as $S = \{I_t \in \mathbb{R}^{N \times 3 \times H \times W}\}_{t=T-k}^{T}$, where $N$ is the number of camera views and $k$ indicates the temporal length till current timestep $T$ respectively. Annotation of input $S$ for end-to-end planning is composed by a set of future waypoints of the ego-vehicle $\psi = \{\phi = (x_t, y_t)\}_{t=1}^{T_p}$, where $T_p = 3s$ is the planning time horizon, and $(x_t, y_t)$ is the BEV location transformed to the ego-vehicle coordinate system at current timestep $T$. Meanwhile, driving command as well as ego-status is also provided. Annotation set $\psi$ is used during training. During prediction, the planned trajectory of ego-vehicle should fit the annotation $\psi$ with minimum L2 errors and collision rate with surrounding agents.

## 3.3 SPARSE PERCEPTION

After extracting the multi-view visual features $F$ from sensor images using the visual encoder He et al. (2016), sparse perception method proposed in Lin et al. (2023) is extended to perform detection, tracking and online mapping simultaneously based on a group of sparse queries, removing the dependence of dense BEV representations widely used in Hu et al. (2023); Jiang et al. (2023).

**Detection and Tracking.** Following the previous sparse perception methods Liu et al. (2023a); Lin et al. (2023), surrounding agents can be represented by a group of instance features $F_a \in \mathbb{R}^{N_a \times C}$ and anchor boxes $B_a \in \mathbb{R}^{N_a \times 11}$ respectively. And each anchor box is denoted as $\{x, y, z, ln(w), ln(h), ln(l), sin(\theta), con(\theta), v_x, v_y, v_z\}$, which contains location, dimension, yaw angle as well as velocity respectively. Taking the visual features $F$, instance features $F_a$ and anchor boxes $B_a$ as input, $N_{dec}$ decoders are adopted to consecutively refine the anchor boxes and update the instance features through deformable aggregation of sample features projected from key points of the anchor box $B_a$. The updated instance features are adopted to predict the classification scores and box offsets respectively. Temporal instance denoising is introduced to improve model stability. As for tracking, following the ID assignment process in Lin et al. (2023), the temporal instances across frames used for advanced detection can be also served as track queries, which remain consistent with unique IDs.

**Online Mapping.** Similarly, we adopt an additional detection branch of same structure for online mapping. Differently, the geometric anchor of each static map element is denoted as $N_p$ points. Therefore, surrounding maps can be represented by a group of map instance features $F_m \in \mathbb{R}^{N_m \times C}$ and anchor polylines $B_m \in \mathbb{R}^{N_m \times N_p \times 2}$.

### 3.4 EGO-ENV HIERARCHICAL INTERACTION

After perceiving the dynamic and static elements existing in the driving scenario in a sparse manner, we continue to perform hierarchical interaction between the ego-vehicle and surrounding agent / map instances. As shown in Fig. 2, the hierarchical interaction module mainly consists of three parts: *Ego-Env Dual Interaction*, *Intention-guided Geometric Attention* and *Coarse-to-Fine Selection*.

**Ego-Env Dual Interaction.** As shown in Fig. 3, a learnable embedding $F_e \in \mathbb{R}^{1 \times C}$ is randomly initialized to serve as ego query, along with an ego anchor box $B_e \in \mathbb{R}^{1 \times 11}$ together to represent the ego-vehicle. Both ego-centric cross attention with surrounding objects $F_o \in \mathbb{R}^{N_o \times C}$ and object-centric self attention are conducted consecutively to capture the mutual information comprehensively. During the attention calculation process, we adopt the decouple attention mechanism proposed in Lin et al. (2023) to combine positional embedding and query feature in a concatenated manner instead of an additive approach, which can effectively retain both semantic and geometric clues for interaction modeling.

**Intention-Guided Geometric Attention.** To enhance the accuracy and explainability of query ranking to facilitate selection, we introduce an ego-centric geometric prior additionally. As shown in Fig. 2, the intention-guided attention module is adopted to assess the importance of surrounding agent and map queries, which mainly consists of three steps: *Response Map Learning*, *Reference Line Generation* and *Interactive Score Fusion*.

Specifically, we use four MLPs to encode the ego-intention respectively, including velocity, acceleration, angular velocity and driving command. And then we concatenate these embeddings to obtain ego-intention features $I_e \in \mathbb{R}^{1 \times C}$, which are further concatenated with the position embeddings $F_p \in \mathbb{R}^{H \times W \times C}$ of a group of pre-defined locations $P \in \mathbb{R}^{H \times W \times 2}$ to cover densely distributed grid cells in the BEV plane. The position of each grid cell is represented as $p = (x, y)$. Finally, the concatenated geometric features are fed to a single SE Hu et al. (2018) block to learn response map $M_r \in \mathbb{R}^{H \times W \times 1}$, which is supervised by the normalized minimum distance from $p$ to the ego future waypoints. The motivation is that the Closest In-Path Vehicle / Stationary are prone to affect the ego-intention, and vice versa.

With the predicted response map $M_r$, we first generate the reference line through row-wise thresholding, which are further used to generate the normalized distance map $M_d$ (See Fig. 2). Then we can obtain the geometric score $S_{geo}$ for each surrounding query by referring to the $M_d$. The reason why we don't get the geometric score from $M_r$ directly is that the imbalanced distribution of ego-intention and future waypoints may lead to the inferior quality of $M_r$.

Finally, as shown in Fig. 4, we perform interactive score fusion through multiplying the attention, geometric and classification scores during the ego-centric cross attention:

$$S_{inter} = \underbrace{Softmax(F_e \odot F_o^T / \sqrt{d_k})}_{S_{attn} \in \mathbb{R}^{N \times 1}} \cdot S_{geo} \cdot S_{cls}, \tag{1}$$

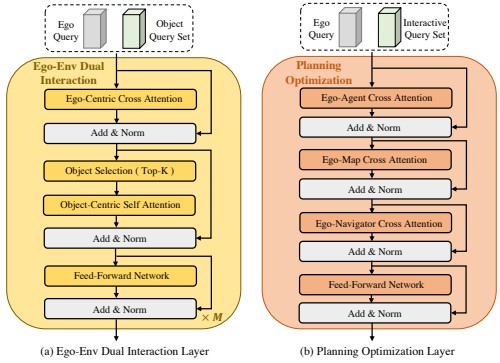

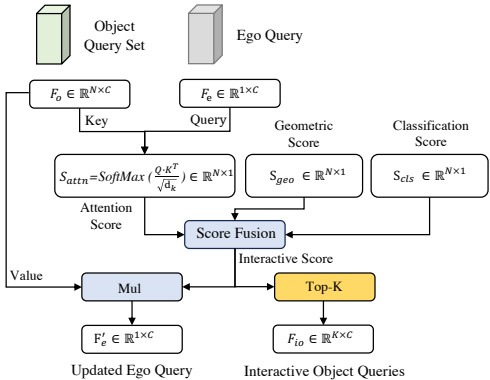

Figure 3: Illustration of the dual interaction layer in the hierarchical interaction module and planning optimization layer in the motion planner module.

Figure 4: Details of the interactive score fusion process in the geometric attended selection step.

where the distance-prior is weighted with the attention score $S_{attn}$ for both interaction and selection. $\odot$ is inner product, $\cdot$ is dot product, and $d_k$ is the channel dimension.

**Coarse-to-Fine Selection.** To capture the interaction information from coarse to fine, we stack $M$ dual-interaction layers in a cascaded manner, where a top-K operation is appended between each two consecutive layers, thus the interactive objects can be gradually selected for latter prediction and planning usages. We claim that only a few interactive objects need to be considered for motion prediction, which are enough yet efficient for ego-centric path planning, instead of all detected agents existing in the driving scene.

## 3.5 ITERATIVE MOTION PLANNER

As shown in Fig. 2, the iterative motion planner is designed to conduct motion prediction for both interactive agents and ego-vehicle, and then optimize the proposal ego-trajectory with both safety and kinematic constrains iteratively.

**Joint Motion Prediction.** With regard to the trajectory prediction, both surrounding agents and ego-vehicle are adopted for motion modeling in a joint decoder, unlike previous works Hu et al. (2023); Ye et al. (2023); Jiang et al. (2023) which neglect the crucial interactions between near agents and ego-vehicle when making motion predictions, especially in the common scenarios like intersections. Another difference is that only the interactive objects $F_{io}$ (CIPV) sparsely selected in the former module are considered, instead of all detected agents in the driving scene which maybe irrelevant to the ego-vehicle planning. As for the joint motion decoder, we prepare three copies of ego query $F_e^{'}$ to indicate different driving intentions (*i.e.,* turn left, turn right and keep forward), which are combined with $F_{io}$ to conduct agent-level self attention and agent-map cross attention respectively. And then we concatenate these output attended features to predict multi-modal trajectories $\tau_a \in \mathbb{R}^{N_a \times K_a \times T_a \times 2}$, $\tau_e \in \mathbb{R}^{N_e \times K_e \times T_e \times 2}$ and classification scores $S_a \in \mathbb{R}^{N_a \times K_a}$, $S_e \in \mathbb{R}^{N_e \times K_e}$ for both agents and ego-vehicle, where $N_e = 3$ is the number of driving command for planning, $K_a = K_e = 6$ are the mode number, $T_a = T_e = 6$ are the future timestamps.

**Planning Optimization.** With the predicted multi-intention and multi-modal trajectories of ego-vehicle, we can select the most probable proposal trajectory with the input driving command and classification score $S_e$. As shown in Fig. 3(b), ego-agent, ego-map and ego-navigator cross attentions are conducted consecutively for planning optimization, where the offsets for each future waypoint are predicted upon the proposal trajectory respectively with several planning constraints proposed in Jiang et al. (2023) to ensure safety.

**Iterative Refinement.** To further promote the stability and performance of the whole end-to-end system, an additional iterative refinement strategy is proposed to continuously update the reference line and distance map $M_d$ with refined ego trajectory as illustrated in Fig. 2, thus ensuring the interaction quality and selection accuracy of interactive queries.

## 3.6 Uncertainty Denoising

Due to the planning-oriented modular design, output uncertainty from each individual module will be inevitably introduced and passed through to the downstream tasks, leading to inferior and fragile system. Under this circumstance, we propose a two-level uncertainty modeling strategy to further stabilize the whole framework. On one hand, **position-level diffusion process** is performed on ground-truth boxes of interactive agents $B_i \in \mathbb{R}^{K \times 11}$ for additional trajectory prediction of noisy agents $B_n = B_i + \Delta B_{pos} \in \mathbb{R}^{G \times K \times 11}$ equipped with $G$ groups of random noises following uniform distributions. $\Delta B_{pos}$ locates within two different ranges of $\{-s, s\}$ and $\{-2s, -s\} \cup \{s, 2s\}$ to indicate positives and negatives respectively, where $s$ indicates the noise scale. This process aims to promote the stability of motion forecasting for interactive agents with uncertain detected positions, scales and velocities. On the other hand, **trajectory-level denoising process** is also introduced for robust offset prediction of proposal trajectory of ego-vehicle in the planning optimization stage. Different from the position diffusion of detection or motion query described above, we apply the random noise to trajectory offsets of ego-vehicle $\Delta B_{traj} \in \mathbb{R}^{G \times T_e \times 2}$, where $s$ depends on the Final Displacement (FD) of ground-truth ego future trajectory.

## 3.7 End-to-End Learning

**Multi-stage Training.** To facilitate the model convergence and training performance, we divide the training process into two stages. In stage-1, the sparse perception, hierarchical interaction and joint motion prediction tasks are trained from scratch to learn sparse scene representation, interaction and motion capability respectively. Note that no selection operation is adopted in stage-1, namely all detected agents are considered for motion forecasting to make full use of annotations. In stage-2, the geometric attention module and the iterative planning optimizer are added to train jointly for overall optimization with uncertainty modeling.

**Loss Functions.** The overall optimization function mainly includes five tasks, where each task can be optimized with both classification and regression losses. The overall loss function for end-to-end training can be formulated as:

$$\mathcal{L} = \mathcal{L}_{det} + \mathcal{L}_{map} + \mathcal{L}_{interact} + \sum_{i=1}^{N}(\mathcal{L}_{motion}^i + \mathcal{L}_{plan}^i), \qquad (2)$$

where $\mathcal{L}_{interact}$ is a combination of binary classification loss and $\mathcal{L}2$ regression loss to learn geometric score, where the positive (interactive) samples are denoted as grid cells with geometric score $S_{geo} \geq 0.9$ (within $3m$ for each future waypoint). An additional regression loss is included in $\mathcal{L}_{plan}$ for ego status prediction, instead of directly using it as input to the planner as Hu et al. (2023); Ye et al. (2023); Jiang et al. (2023), which will lead to information leakage as proven in Li et al. (2024). Meanwhile, vectorized planning constrains identified in Jiang et al. (2023) such as collision, overstepping and direction are also included in $\mathcal{L}_{plan}$ for regularization. $N$ is the number of motion planning stages.

# 4 Experiments

## 4.1 Datasets and Setup

Our experiments are conducted on the challenging public nuScenes Caesar et al. (2020) dataset, which contains 1000 driving scenes lasting 20 seconds respectively. Over 1.4M 3D bounding boxes of 23 categories are provided in total, which are annotated at 2Hz. Following the conventions Hu et al. (2023); Jiang et al. (2023), Collision Rate (%) and L2 Displacement Error (DE) $(m)$ are adopted to measure the open-loop planning performance. Besides, to study the effect of various perception encoders, we evaluate the 3D object detection and online mapping results using mAP and NDS metrics respectively.

## 4.2 Implementation Details

DiFSD plans a $3s$ future trajectory of ego-vehicle with $2s$ history information as input. Our DiFSD has two variants, namely DiFSD-B and DiFSD-S. As for DiFSD-S, both sparse perception version

Table 1: Open-loop planning evaluation results on the nuScenes **val** dataset. $*$ denotes multi-modality fusion method. † indicates evaluation with official checkpoint. ‡ indicates using evaluation protocol proposed in Zhai et al. (2023); Li et al. (2024).

| Method | Backbone | L2 (m) ↓ | | | | Collision (%) ↓ | | | | Latency (ms) ↓ | FPS ↑ |
|---|---|---|---|---|---|---|---|---|---|---|---|
| | | 1s | 2s | 3s | Avg. | 1s | 2s | 3s | Avg. | | |
| ST-P3 | EfficientNet-b4 | 1.33 | 2.11 | 2.90 | 2.11 | 0.23 | 0.62 | 1.27 | 0.71 | 628.3 | 1.6 |
| FusionAD$^*$ | R101+SECOND | - | - | - | 0.81 | 0.02 | 0.08 | 0.27 | 0.12 | - | - |
| UniAD | ResNet101-DCN | 0.48 | 0.96 | 1.65 | 1.03 | 0.05 | 0.17 | 0.71 | 0.31 | 555.6 | 1.8 |
| VAD | ResNet50 | 0.41 | 0.70 | 1.05 | 0.72 | 0.07 | 0.17 | 0.41 | 0.22 | 224.3 | 4.5 |
| SparseDrive-S † | ResNet50 | 0.30 | 0.58 | 0.95 | 0.61 | 0.47 | 0.47 | 0.69 | 0.54 | 111.1 | 9.0 |
| DiFSD-S (Dense) | ResNet50 | **0.16** | **0.33** | **0.59** | **0.35** | **0.00** | **0.04** | **0.18** | **0.07** | **67.7** | **14.8** |
| UniAD‡ | ResNet101-DCN | 0.45 | 0.70 | 1.04 | 0.73 | 0.62 | 0.58 | 0.63 | 0.61 | 555.6 | 1.8 |
| VAD ‡ | ResNet50 | 0.41 | 0.70 | 1.05 | 0.72 | 0.03 | 0.19 | 0.43 | 0.21 | 224.3 | 4.5 |
| SparseDrive-S ‡ | ResNet50 | 0.30 | 0.58 | 0.95 | 0.61 | 0.01 | 0.05 | 0.23 | 0.10 | 111.1 | 9.0 |
| SparseDrive-B ‡ | ResNet101 | 0.29 | 0.55 | 0.91 | 0.58 | 0.01 | **0.02** | **0.13** | **0.06** | 137.0 | 7.3 |
| DiFSD-S (Dense)‡ | ResNet50 | 0.16 | 0.33 | 0.59 | 0.35 | 0.03 | 0.07 | 0.21 | 0.10 | **67.7** | **14.8** |
| DiFSD-S (Sparse)‡ | ResNet50 | **0.15** | 0.31 | 0.56 | 0.33 | **0.00** | 0.06 | 0.19 | 0.08 | 93.7 | 10.7 |
| DiFSD-B (Sparse)‡ | ResNet101 | **0.15** | **0.30** | **0.54** | **0.32** | **0.00** | 0.04 | 0.15 | **0.06** | 119.6 | 8.4 |

Table 2: Comparison of perception results (3D detection and online mapping) of state-of-the-art perception or end-to-end methods on nuScenes **val** dataset. †: Reproduced with official checkpoint. $*$ indicates to use pre-trained weights from the nuImage dataset.

| Method | Backbone | Sparse | mAP ↑ | NDS ↑ |
|---|---|---|---|---|
| BEVFormer Li et al. (2022b) | ResNet101-DCN | ✗ | 41.6 | 51.7 |
| Sparse4Dv3 Lin et al. (2023) | ResNet101$^*$ | ✓ | 53.7 | 62.3 |
| UniAD Hu et al. (2023) | ResNet101-DCN | ✗ | 38.0 | 49.8 |
| VAD$^†$ Jiang et al. (2023) | ResNet50 | ✗ | 27.3 | 39.7 |
| SparseDrive-S Sun et al. (2024) | ResNet50 | ✓ | 41.8 | 52.5 |
| SparseDrive-B Sun et al. (2024) | ResNet101$^*$ | ✓ | 49.6 | 58.8 |
| DiFSD-S | ResNet50 | ✗ | 32.8 | 45.8 |
| DiFSD-S | ResNet50 | ✓ | 41.0 | 52.8 |
| DiFSD-B | ResNet101$^*$ | ✓ | 49.6 | 58.9 |

| Method | $AP_{ped}$ ↑ | $AP_{divider}$ ↑ | $AP_{boundary}$ ↑ | mAP ↑ |
|---|---|---|---|---|
| VectorMapNet Liu et al. (2023b) | 36.1 | 47.3 | 39.3 | 40.9 |
| MapTR Liao et al. (2022) | 56.2 | 59.8 | 60.1 | 58.7 |
| VAD$^†$ Jiang et al. (2023) | 40.6 | 51.5 | 50.6 | 47.6 |
| SparseDrive-S Sun et al. (2024) | 49.9 | 57.0 | 58.4 | 55.1 |
| SparseDrive-B Sun et al. (2024) | 53.2 | 56.3 | 59.1 | 56.2 |
| DiFSD-S (Dense) | 46.7 | 54.3 | 56.0 | 52.3 |
| DiFSD-S (Sparse) | 54.9 | 55.7 | 57.3 | 56.0 |
| DiFSD-B (Sparse) | 52.3 | 58.2 | 59.3 | 56.6 |

(a) 3D detection results.    (b) Online mapping results

DiFSD-S (Sparse) and dense BEV perception version DiFSD-S (Dense) are all implemented for comparison. ResNet50 He et al. (2016) is adopted as the default backbone network for visual encoding. The perception range is set to $60m \times 30m$ longitudinally and laterally. Input image size of DiFSD-S is resized to $640 \times 360$. For DiFSD-S (Dense), the default number of BEV query, map query, agent query is $100 \times 100$, $100 \times 20$ and 300, respectively. For DiFSD-S (Sparse), $N_{dec}$ is 6, $N_a$ is 900 and $N_m$ is 100 respectively. Each map element contains 20 map points. The feature dimension $C$ is 256. The noise scale $s$ is set to 2.0 and $0.2 \times$FD for motion and planning respectively. $G$ is set to 3. DiFSD-B has larger input image resolution ($1280 \times 720$) and backbone network (ResNet101). We use AdamW Loshchilov & Hutter (2017) optimizer and Cosine Annealing Loshchilov & Hutter (2016) scheduler to train DiFSD with weight decay 0.01 and initial learning rate $2 \times 10^{-4}$. DiFSD is trained for 48 epochs in stage-1 and 20 epochs in stage-2, running on 8 NVIDIA Tesla A100 GPUs with batch size 1 per GPU.

### 4.3 MAIN RESULTS

As show in Tab. 1, DiFSD shows great advantages in both performance and efficiency compared with previous works, including either visual-based or multi-modality based methods. *On one hand, DiFSD-S achieves the minimum L2 error even with lightweight visual backbone and inferior dense perception encoder.* Specifically, compared with BEVFormer-based end-to-end methods Hu et al. (2023); Jiang et al. (2023), DiFSD-S (Dense) reduces the average L2 error by a great margin ($0.68m$ and $0.37m$, separately), while significantly reducing the average collision rates by 77% and 68% respectively. Equipped with deeper visual backbone and advanced sparse perception, the average L2 error and collision rates can be further reduced to $0.32m$ and to $0.06\%$ respectively. *Notably, we are the first to achieve 0% collision rate on 1s.* On the other hand, benefiting from the ego-centric hierarchical interaction, only sparse interactive agents (2%) are considered for motion planning. Hence, DiFSD-S can achieve great efficiency with 14.8 FPS, $8.2\times$ and $3.3\times$ faster than UniAD Hu et al. (2023) and VAD Jiang et al. (2023) respectively.

### 4.4 ABLATION STUDY

We conduct extensive experiments to study the effectiveness and necessity of each design choice proposed in our DiFSD. We use DiFSD-S as the default model for ablation.

Table 3: Effect of ego-centric query selector and geometric prior.

| Object Selection | Geometric Attention | Planning L2 (m) ↓ | | | | Planning Coll. (%) ↓ | | | |
|---|---|---|---|---|---|---|---|---|---|
| | | 1s | 2s | 3s | Avg. | 1s | 2s | 3s | Avg. |
| 100% | ✗ | 0.27 | 0.47 | 0.74 | 0.49 | 0.10 | 0.21 | 0.37 | 0.22 |
| Random (5%) | ✗ | 0.28 | 0.49 | 0.79 | 0.52 | 0.08 | 0.17 | 0.38 | 0.21 |
| Random (2%) | ✗ | 0.33 | 0.57 | 0.87 | 0.59 | 0.18 | 0.30 | 0.51 | 0.33 |
| 0% | ✗ | 2.25 | 3.75 | 5.26 | 3.75 | 2.82 | 5.42 | 6.39 | 4.88 |
| Attn (5%) | ✗ | 0.16 | 0.34 | 0.63 | 0.38 | 0.07 | 0.09 | 0.31 | 0.16 |
| Attn (2%) | ✗ | 0.16 | 0.34 | 0.61 | 0.37 | 0.08 | 0.11 | 0.27 | 0.15 |
| Attn (2%) | Random | 0.17 | 0.36 | 0.67 | 0.40 | 0.07 | 0.10 | 0.34 | 0.17 |
| Attn (2%) | GroundTruth | 0.14 | 0.23 | 0.33 | 0.23 | 0.07 | 0.08 | 0.10 | 0.07 |
| Attn (2%) | ✓ | 0.16 | 0.33 | 0.59 | **0.35** | 0.00 | 0.04 | 0.18 | **0.07** |

Table 4: Ablation for designs in the hierarchical interaction. "DI" means dual interaction; "GA" means geometric attention; "CFS" means coarse-to-fine selection.

| DI | GA | CFS | Planning L2 (m) ↓ | | | | Planning Coll. (%) ↓ | | | |
|---|---|---|---|---|---|---|---|---|---|---|
| | | | 1s | 2s | 3s | Avg. | 1s | 2s | 3s | Avg. |
| ✗ | ✓ | ✓ | 0.18 | 0.35 | 0.62 | 0.38 | 0.09 | 0.12 | 0.23 | 0.14 |
| ✓ | ✗ | ✓ | 0.16 | 0.34 | 0.61 | 0.37 | 0.08 | 0.11 | 0.27 | 0.15 |
| ✓ | ✓ | ✗ | 0.16 | 0.33 | 0.59 | 0.36 | 0.09 | 0.11 | 0.25 | 0.15 |
| ✓ | ✓ | ✓ | 0.16 | 0.33 | 0.59 | **0.35** | 0.00 | 0.04 | 0.18 | **0.07** |

Table 5: Ablation for designs in the motion planner. "JMP": joint motion prediction; "PO": planning optimization; "IR": iterative refinement. "UD": uncertainty denoising.

| ID | JMP | PO | IR | UD | Planning L2 (m) ↓ | | | | Planning Coll. (%) ↓ | | | |
|---|---|---|---|---|---|---|---|---|---|---|---|---|
| | | | | | 1s | 2s | 3s | Avg. | 1s | 2s | 3s | Avg. |
| 1 | ✓ | ✗ | ✗ | ✓ | 0.23 | 0.48 | 0.83 | 0.51 | 0.08 | 0.13 | 0.35 | 0.18 |
| 2 | ✓ | ✓ | ✗ | ✓ | 0.16 | 0.33 | 0.61 | 0.37 | 0.01 | 0.08 | 0.23 | 0.11 |
| 3 | ✓ | ✓ | ✓ | ✗ | 0.16 | 0.34 | 0.64 | 0.38 | 0.07 | 0.07 | 0.17 | 0.10 |
| 4 | ✓ | ✓ | ✓ | ✓ | 0.16 | 0.33 | 0.59 | **0.35** | 0.00 | 0.04 | 0.18 | **0.07** |

**Effect of Sparse Perception.** In addition to BEV-perception based end-to-end methods Hu et al. (2023); Jiang et al. (2023), recent end-to-end planning method Sun et al. (2024) resorts to the sparse perception fashion to provide advanced 3D detection and online mapping results with high efficiency. To study the significance of advanced perception encoders for ego-planning, we compare the perception performance of various end-to-end methods as shown in Tab. 2. With sparse perception encoder Lin et al. (2023), the performance of 3D object detection and online mapping can be greatly improved (10.6 NDS and 7.5 mAP, respectively) compared with dense BEV-based perception paradigm Li et al. (2022b). And the end-to-end planner Sun et al. (2024) equipped with the advanced perception encoder can consistently boost the planning performance as shown in Tab. 1. Therefore, the perception performance is essential for the end-to-end planner, which decides the planning upper-bound and provides rich clues of surrounding environment including both dynamic and static elements.

**Necessity of Geometric Prior.** We claim that only interactive agent and map queries are significant for ego-vehicle planning, where the Closest In-Path Vehicle as well as Stationary (CIPV / CIPS) are more likely to interact with the ego-vehicle. To verify the necessity of such geometric prior, we conduct exhaustive ablations of the ego-centric query selector as show in Tab. 3. Without ego-centric selection, fewer objects randomly selected can result in worse planning results. While using the ego-centric cross attention, only 2% of surrounding queries are enough for achieving convincing planning performance, instead of considering all existing dynamic/static elements. Besides, introducing the geometric prior through attention can dramatically reduce the L2 error and collision rate by 8% and 42% respectively. Meanwhile, when utilizing the ground-truth geometric score for upper-limit evaluation, we can obtain the extremely lower average L2 error and collision rate ($0.23m$ and 0.07% respectively). Undoubtedly, the proposed ego-centric selector equipped with geometric attention is nontrivial for efficient interaction and motion planner.

**Effect of designs in Hierarchical Interaction.** Tab. 4 shows the effectiveness of our elaborate designs in the hierarchical interaction module, which contains three main designs such as Dual Interaction (DI), Geometric Attention (GA) and Coarse-to-Fine Selection (CFS). DI models both ego-centric and object-centric interactions respectively, which improves the planning performance greatly as expected. GA facilitates the query selection process as discussed in Tab. 3, which reduces the collision rate by a great margin (42%). And CFS contributes to the interaction modeling quality through hierarchical receptive fields from global to local. All of these three designs combined together can achieve overall convincing planning performance.

**Effect of designs in Motion Planner.** As for motion planner in DiFSD, Joint Motion Prediction (JMP), Planning Optimization (PO) as well as Iterative Refinement (IR) makes up the planning pipeline of ego-vehicle. Besides, Uncertain Denoising (UD) contributes to the system stability and

Table 6: Module runtime statistics.The inference speed is measured for DiFSD-S on one NVIDIA Tesla A100 GPU. Different perception fashions are both considered for comparisons.

| Module | Dense BEV Fashion | | Fully Sparse Fashion | |
|---|---|---|---|---|
| | Latency (*ms*) | Proportion (%) | Latency (*ms*) | Proportion (%) |
| Backbone | 8.4 | 12.4 | 11.0 | 11.8 |
| Perception | 34.4 | 50.8 | 54.9 | 58.6 |
| Hierarchical Interaction | 17.0 | 25.1 | 19.9 | 21.2 |
| Joint Motion Prediction | 4.5 | 6.6 | 4.5 | 4.8 |
| Planning Optimization | 3.4 | 5.1 | 3.4 | 3.6 |
| Total | 67.7 | 100 | 93.7 | 100 |

training convergence. Tab. 5 explores the effect of each design exhaustively. ID-1 indicates evaluating the proposal trajectory of ego-vehicle predicted together with interactive agents, which achieves competitive L2 error but is easier to collide with surrounding agents. ID-2 improves the collision rate greatly by 38.9% with the help of PO and planning constraints Jiang et al. (2023) during training phase. ID-4 emphasizes the importance of IR in improving the quality of ego-planning trajectory (average 5.4% L2 error and 36.3% collision rate reduction respectively). ID-3 reflects the benefit of UD used for end-to-end training compared with ID-4.

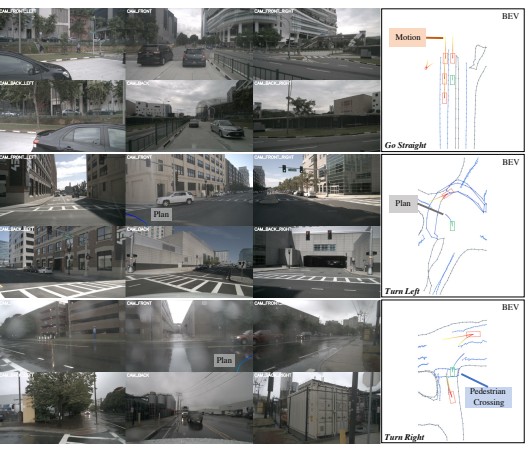

Figure 5: Qualitative results of DiFSD. DiFSD outputs planning results based on hierarchical interaction and joint motion of sparse interactive agents without considering other irrelevant objects. We omit the map selection results for clarity of road structure details.

**Runtime of each module.** As shown in Tab. 6, visual backbone and sparse perception occupy the most of the runtime (70.4%) for feature extraction and scene understanding. Hierarchical interaction also takes a significant part (21.2%) for interaction modeling and interactive query selection. Thanks to the sparse representation and ego-centric interaction module, the motion planner only consumes 7.9*ms* to plan the future ego-trajectory (8.4% in total).

### 4.5 QUALITATIVE RESULTS

We visualize the motion trajectories of interactive agents as well as planning results of DiFSD as illustrated in Fig. 5. Both surrounding camera images and prediction results on BEV are provided accordingly. Besides, we also project the planning trajectories to the front-view camera image. Only the top-3 trajectories of selected agents interacting with ego-vehicle are visualized for better understanding of DiFSD motivation. DiFSD outputs planning results based on the fully sparse representation in an end-to-end manner, not requiring any dense interaction and redundant motion modeling, let alone hand-crafted post-processing.

## 5 CONCLUSION

In this paper, we propose a fully sparse paradigm for end-to-end self-driving in an ego-centric manner, termed as DiFSD. DiFSD revisits the human driving behavior and conducts hierarchical interaction based on sparse representation and perception results. Only interactive agents are considered for joint motion prediction with the ego-vehicle. Iterative planning optimization strategy contributes to the driving safety with high efficiency. Besides, uncertainty modeling is conducted to improve the stability of end-to-end system. Extensive ablations and comparisons reveal the superiority and great potential of our ego-centric fully sparse paradigm for future research.

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
