# A EVALUATION METRICS

**Perception.** The evaluation for detection and tracking follows standard evaluation protocols Caesar et al. (2020). For detection, we use mean Average Precision(mAP), mean Average Error of Translation(mATE), Scale(mASE), Orientation(mAOE), Velocity(mAVE), Attribute(mAAE) and nuScenes Detection Score(NDS) to evaluate the model performance. For online mapping, we calculate the Average Precision(AP) of three map classes: lane divider, pedestrian crossing and road boundary, then average across all classes to get mean Average Precision(mAP).

**Planning.** We adopt commonly used L2 error and collision rate to evaluate the planning performance. The evaluation of L2 error is aligned with VAD Jiang et al. (2023). For collision rate, there are two drawbacks in previous Hu et al. (2023); Jiang et al. (2023) implementation, resulting in inaccurate evaluation in planning performance. On one hand, previous benchmark convert obstacle bounding boxes into occupancy map with a grid size of 0.5m, resulting in false collisions in certain cases, e.g. ego vehicle approaches obstacles that smaller than a single occupancy map pixel Zhai et al. (2023). (2) The heading of ego vehicle is not considered and assumed to remain unchanged Li et al. (2024). To accurately evaluate the planning performance, we account for the changes in ego heading by estimating the yaw angle through trajectory points, and assess the presence of a collision by examining the overlap between the bounding boxes of ego vehicle and obstacles. We reproduce the planning results on our benchmark with official checkpoints Hu et al. (2023); Jiang et al. (2023) for a fair comparison.

# B MORE ABLATION STUDY

**Necessity and Order of Object Selection.** Tab. 1 studies the necessity of agent and map selection during the ego-centric hierarchical interaction. We can observe that agent selection contributes more than the map selection, especially in the driving safety. And both of agent and map interactions are conducted in a cascaded order is inferior than the parallel manner, where the updated ego query from parallel outputs are concatenated for joint motion prediction.

Table 1: Effect of agent and map selection as well as interaction order in the hierarchical interaction module.

| Agent Selection | Map Selection | Cascade | Parallel | Planning L2 (m) ↓ | | | | Planning Coll. (%) ↓ | | | |
|:---:|:---:|:---:|:---:|:---:|:---:|:---:|:---:|:---:|:---:|:---:|:---:|
| | | | | 1s | 2s | 3s | Avg. | 1s | 2s | 3s | Avg. |
| ✓ | ✗ | - | - | 0.16 | 0.34 | 0.64 | 0.38 | 0.03 | 0.05 | 0.22 | 0.10 |
| ✗ | ✓ | - | - | 0.17 | 0.35 | 0.63 | 0.38 | 0.02 | 0.06 | 0.28 | 0.12 |
| ✓ | ✓ | ✓ | - | 0.16 | 0.34 | 0.62 | 0.37 | 0.05 | 0.07 | 0.30 | 0.14 |
| ✓ | ✓ | - | ✓ | 0.16 | 0.33 | 0.59 | **0.35** | 0.00 | 0.04 | 0.18 | **0.07** |

**Effect of Interactive Score Fusion.** During the ego-centric query selection, both geometric and classification scores are considered to ensure that the selected closest in-path queries are true positive agents or maps, which are adopted for motion planner. Tab. 2 shows the effect of three types of scores used for query ranking, namely attention, geometry and confidence scores. As described above, interactive score $S_{inter}$ obtained by multiplying these three scores can achieve the best selection quality and planning performance. $S_{inter}$ without confidence score fails to distinguish between background and foreground queries, resulting in inferior performance.

Table 2: Effect of interactive score fusion process in the geometric attended selection step.

| Attention Score | Geometric Score | Classification Score | Planning L2 (m) ↓ | | | | Planning Coll. (%) ↓ | | | |
|:---:|:---:|:---:|:---:|:---:|:---:|:---:|:---:|:---:|:---:|:---:|
| | | | 1s | 2s | 3s | Avg. | 1s | 2s | 3s | Avg. |
| ✓ | ✗ | ✗ | 0.18 | 0.36 | 0.66 | 0.39 | 0.09 | 0.11 | 0.28 | 0.16 |
| ✓ | ✓ | ✗ | 0.17 | 0.35 | 0.65 | 0.38 | 0.01 | 0.07 | 0.24 | 0.11 |
| ✓ | ✓ | ✓ | 0.16 | 0.33 | 0.59 | **0.35** | 0.00 | 0.04 | 0.18 | **0.07** |

**Effect of Iterative Refinement stages.** We continue to study the number of refinement stages in Tab. 3. We can observe that our DiFSD can obtain superior planning performance with one addi-

tional refinement stage (36.3% collision rate reduction), which becomes saturated when introducing more stages. Hence, two-stage interacted motion planner is enough for achieving convincing results.

Table 3: Ablation for number of iterative refinement stages.

| Number of stages | Planning L2 (m) ↓ | | | | Planning Coll. (%) ↓ | | | |
|---|---|---|---|---|---|---|---|---|
| | 1s | 2s | 3s | Avg. | 1s | 2s | 3s | Avg. |
| 1 | 0.16 | 0.33 | 0.61 | 0.37 | 0.01 | 0.08 | 0.23 | 0.11 |
| 2 | 0.16 | 0.33 | 0.59 | **0.35** | 0.00 | 0.04 | 0.18 | **0.07** |
| 3 | 0.16 | 0.33 | 0.60 | 0.36 | 0.01 | 0.40 | 0.22 | 0.09 |
| 4 | 0.16 | 0.33 | 0.61 | 0.36 | 0.00 | 0.04 | 0.20 | 0.08 |

**Effect of Uncertainty Denoising.** We also validate the effectiveness of uncertainty denoising strategy including position-level motion diffusion and trajectory-level planning denoising. As shown in Tab. 4, motion diffusion can improve the prediction stability with uncertain agent positions, while the planning denoising can also strengthen the trajectory regression precision of ego-vehicle.

Table 4: Ablation for uncertainty denoising procedure.

| Position Diffusion | Trajectory Denoising | Planning L2 (m) ↓ | | | | Planning Coll. (%) ↓ | | | |
|---|---|---|---|---|---|---|---|---|---|
| | | 1s | 2s | 3s | Avg. | 1s | 2s | 3s | Avg. |
| ✗ | ✗ | 0.16 | 0.34 | 0.64 | 0.38 | 0.07 | 0.07 | 0.17 | 0.10 |
| ✓ | ✗ | 0.16 | 0.34 | 0.63 | 0.37 | 0.02 | 0.04 | 0.15 | 0.07 |
| ✓ | ✓ | 0.16 | 0.33 | 0.59 | **0.35** | 0.00 | 0.04 | 0.18 | **0.07** |

## C  ANALYSIS & DISCUSSION

The GroundTruth future state distribution of ego-vehicle on nuScenes validation set is illustrated in Fig. 1, which is calculated with fixed time interval ($1s$) between consecutive predicted waypoints. And we also compare the output ego-state distribution of different popular end-to-end methods based on planned trajectories respectively, as show in Fig. 2. We can observe that without ego-centric design, the optimized end-to-end model is unable to handle various emergencies appearing in the driving scenarios, where the absolute values of $\Delta v$ and $\Delta a$ are larger than normal situations. Under this circumstance, the output planned trajectories cannot conform to the expert routes as expected. However, our DiFSD performs consistently better in planning the future ego states with variable speed and acceleration, owing to the ego-centric hierarchical interaction and selection mechanism, thus the iterative motion planner can focus on the interactive agents rather than irrelevant objects.

## D  VISUALIZATION

As show in Fig. 3, 4 and 5, we provide additional visualization results to illustrate the generalizability of DiFSD on various driving scenarios under different commands.

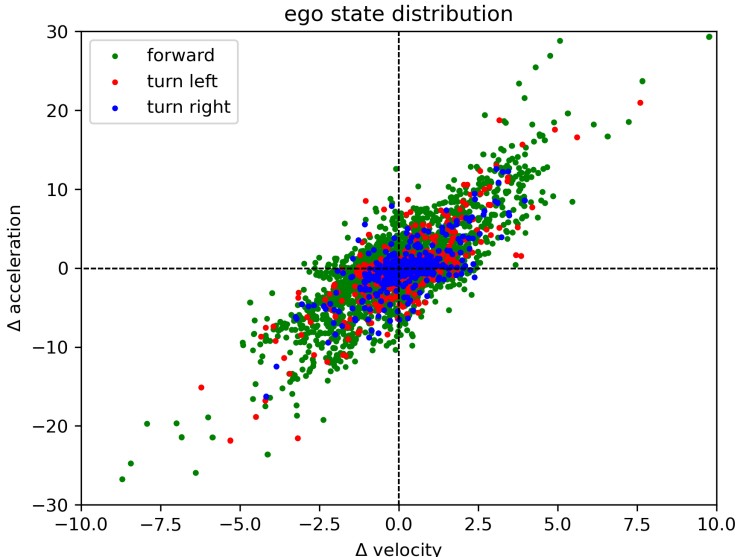

Figure 1: Distribution of GroundTruth future ego states ($\Delta v$ vs. $\Delta a$) with different driving commands on the nuScenes **val** set.

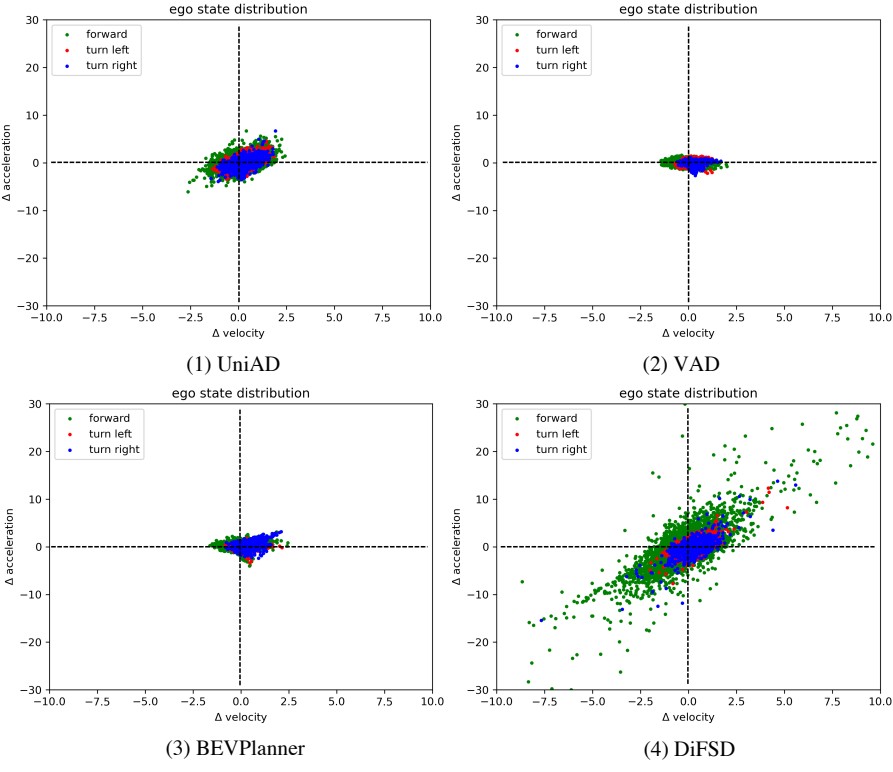

Figure 2: Comparison of predicted future ego states of different end-to-end methods on the validation set of nuScenes.

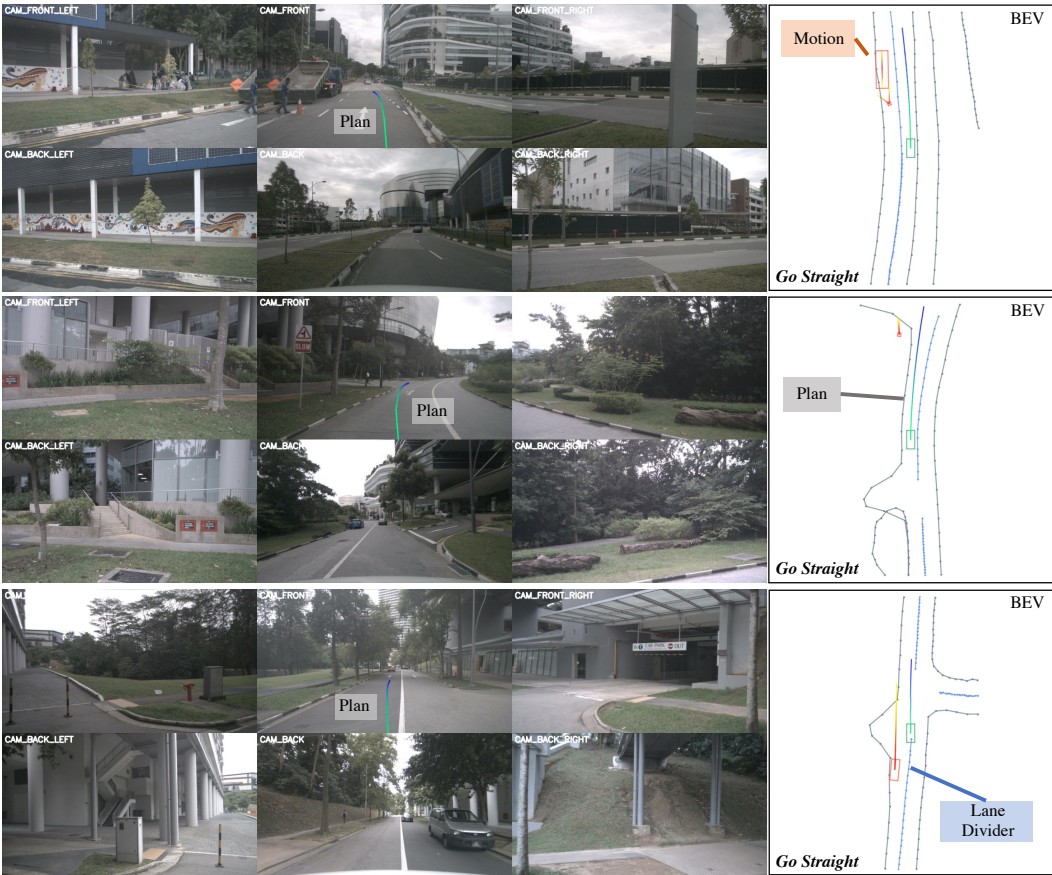

Figure 3: Qualitative results of DiFSD under "*Go Straight*" driving command in interactive scenes. In the first row, the pedestrian and the construction vehicle are selected as the closest in-path agents for motion prediction and interactive planning, thus DiFSD adjusts the planned trajectory from afar to avoid a collision. In the second row, DiFSD notices the pedestrian in the distance and plans the future trajectory taking the pedestrian intention into consideration. In the third row, DiFSD completes interactive decision-making in the "Cut-in" scenario, and outputs the planned trajectory constrained by the lane divider.

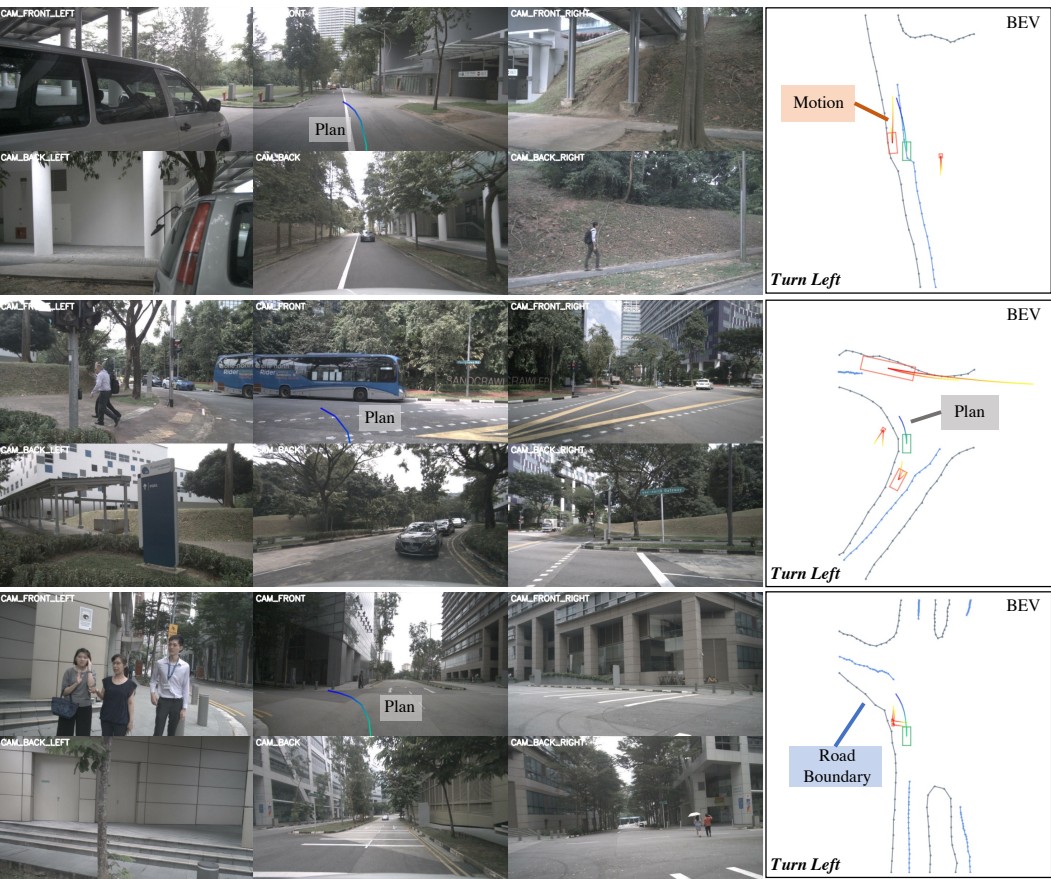

Figure 4: Qualitative results of DiFSD under "*Turn Left*" driving command in diverse scenarios. In the first scenario, DiFSD makes an "Overtaking" decision from the ride side of the front vehicle, considering the motions of both target vehicle and neighboring pedestrian to ensure driving safety. In the latter two intersection scenarios, DiFSD detects the pedestrians waiting at the crossing and the opposite bus passing the intersection, then decelerates to make a turning decision.

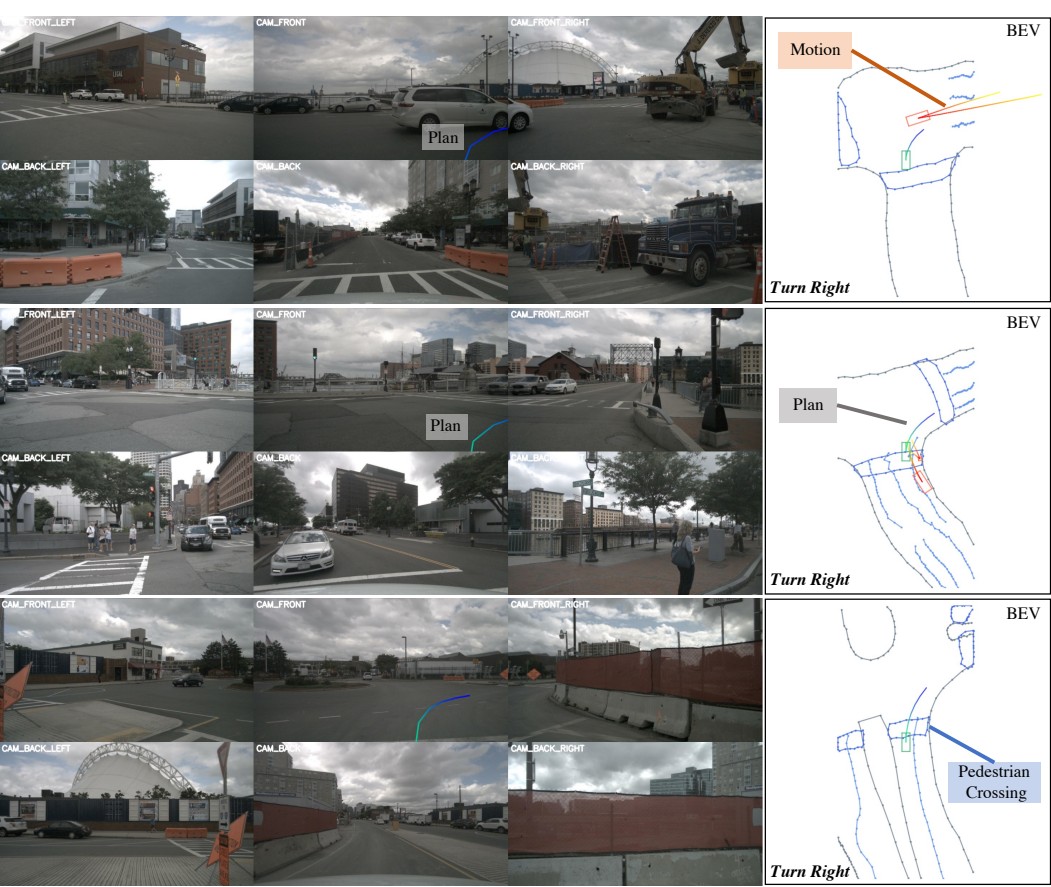

Figure 5: Qualitative results of DiFSD under "*Turn Right*" driving command at both interactive and non-interactive intersections. Joint motion prediction of agents and ego-vehicle is essential for DiFSD especially in the turning scenarios at interactions. The first two rows illustrate the interactive scenarios either inside and outside the intersection. And the last row presents a non-interactive intersection where DiFSD plans the future trajectory merely based on the detected pedestrian crossing.