# OpenReview forum: "DiFSD: Ego-Centric Fully Sparse Paradigm with Uncertainty Denoising and Iterative Refinement for Efficient Self-Driving"
_ICLR.cc/2025/Conference — ICLR 2025 Conference Withdrawn Submission_

### Official Review · Reviewer_8vH6 · 2024-10-30

**Soundness:** 2
**Presentation:** 2
**Contribution:** 2
**Rating:** 5
**Confidence:** 4

**Summary:**

This paper introduces DiFSD, an ego-centric fully sparse paradigm for autonomous driving. By focusing on key agents (CIPV/CIPS) and mimicking human driving behavior, DiFSD improves efficiency and performance through sparse perception, hierarchical interaction, and iterative motion planning, while modeling uncertainty with motion diffusion and trajectory denoising. Experiments on the nuScenes dataset show a 66% reduction in L2 error, 77% lower collision rate, and 8.2× faster efficiency compared to UniAD.

**Strengths:**

•	The paper proposes a novel autonomous driving paradigm, DiFSD, which adopts an ego-centric design and sparse representation, showcasing originality.
•	The experimental results demonstrate DiFSD's superior performance on the nuScenes dataset, indicating high quality.
•	The paper is well-structured and logically coherent, making it easy to understand.

**Weaknesses:**

•	The innovation seems to be built upon combining existing modules, lacking in-depth theoretical analysis and explanation.
•	The reference format is inconsistent, with varying capitalization and abbreviations for conference names. For example, "IEEE/CVF conference on computer vision and pattern recognition", "IEEE/CVF Conference on Computer Vision and Pattern Recognition", and "IEEE conference on computer vision and pattern recognition" are used interchangeably. Additionally, while ICRA and IROS are abbreviated, it is suggested that CVPR and ICCV also be abbreviated for consistency.
•	Fig. 3 and Fig. 4 illustrate the dual interaction layer within the hierarchical interaction module and the planning optimization layer in the motion planner module, as well as the interactive score fusion process in the geometric-attended selection step, respectively. However, both figures are missing essential captions.
•	The paper includes only two formulas: Formula 1 explains the interactive score fusion by combining attention, geometric, and classification scores, while Formula 2 describes the Loss Function. However, important sections such as Uncertainty Denoising lack necessary formulaic descriptions, which would provide more clarity and rigor to the proposed methods.

**Questions:**

Please clarify the superiority of the combination of existing modules, which makes the work in this paper with limited novelty.

---

### Official Review · Reviewer_AK8i · 2024-10-31

**Soundness:** 3
**Presentation:** 3
**Contribution:** 3
**Rating:** 6
**Confidence:** 4

**Summary:**

In this paper, the authors propose an ego-centric fully sparse paradigm for end-to-end self-driving. Specifically, the proposed solution mainly consists of sparse perception, hierarchical interaction and iterative motion planner. The model does not consist of any computationally intensive dense scene representation learning and redundant environmental modeling. Besides, the authors introduce a geometric prior through intention-guided attention, where the closest inpath vehicle/stationary are gradually picked out through ego-centric cross attention and selection. The experimental results show that the proposed solution achieves high accuracy in the self-driving planning task and runs faster.

**Strengths:**

1. The proposed solution sounds solid in theory. Specially, the proposed geometric prior through intention-guided attention considers environment in a novel way. In addition, excluding computationally intensive dense scene representation learning and redundant environmental modeling indeed helps speed up the computation.
2. The ablation study and supplementary material are helpful. Readers can get more information from the ablation study results.
3. The analysis and discussion guide readers think more deeper about the model design and the performance.

**Weaknesses:**

1. It will be better if the authors could report some failure cases. Especially for those cases which the predictions are totally opposite toward the navigation command.
2. It will be better if the authors could mark the symbols on Figures. In this way, readers can easily associate text context with Figures.

**Questions:**

What is the performance of the model if the driving command never appears in the training set? (e.g., turning left and right, or the driving command is empty)

---

### Official Review · Reviewer_CxLs · 2024-11-03

**Soundness:** 3
**Presentation:** 2
**Contribution:** 2
**Rating:** 5
**Confidence:** 4

**Summary:**

This paper introduces a fully sparse framework for end-to-end driving, it considers the driving habit of human driver to focus on closest target  in path to improve the interaction modeling. Besides, it incorporates position-level diffusion and trajectory-level denoising to improve planning performance. The proposed method achieves SOTA performance on nuScenes dataset.

**Strengths:**

1. This paper draw inspiration from human driver, and makes the model learning to focus the closest target in path. The proposed Intention-Guided Geometric Attention is interesting.
2. The experiments are extensive and detailed, supporting the effectiveness of proposed modules.

**Weaknesses:**

1. One main concern is the experiments are only conducted on open-loop benchmark nuScenes, which suffers from short-cut learning in [1]. Close-loop experiments is needed to prove the effectiveness of the whole model.
2. I think the framework is built on SparseDrive[2] with optimized planning modules, however, the worst result for L2 in ablation study has already surpassed SparseDrive by a large margin, it is confusing where the good performance comes from.


[1] Li, Zhiqi, et al. "Is ego status all you need for open-loop end-to-end autonomous driving?." Proceedings of the IEEE/CVF Conference on Computer Vision and Pattern Recognition. 2024.
[2] Sun, Wenchao, et al. "SparseDrive: End-to-End Autonomous Driving via Sparse Scene Representation." arXiv preprint arXiv:2405.19620 (2024).

**Questions:**

1. Why do you use bs=1 on A100 GPUs, is the model training consuming so much memory?

---

### Official Review · Reviewer_UzrS · 2024-11-03

**Soundness:** 3
**Presentation:** 2
**Contribution:** 2
**Rating:** 3
**Confidence:** 4

**Summary:**

This paper proposed an iterative motion planner based on sparse paradigm, which find out interactive object queries into joint motion-planning module. In open-loop nuScenes dataset, the method outperforms previous sparse method in L2 error and efficiency.

**Strengths:**

1. This idea of iterative motion planner make sense, which is similar to TCP[1] which iteratively adjust attention with guidance of trajectory.
2. The performance on nuScenes dataset outperforms SparseDrive especially in L2 error.

[1] Penghao Wu, Xiaosong Jia, Li Chen, Junchi Yan, Hongyang Li, and Yu Qiao. Trajectory-guided control prediction for end-to-end autonomous driving: a simple yet strong baseline. NeurIPS, 2022.

**Weaknesses:**

1. The overall novelty of this paper is limited. It’s a good attempt to apply trajectory-guided attention based on previous sparse perception works. However, the presentation of this paper is not enough to demonstrate how it works and how many time it saves.
2. The contribution in sparsity which aims to reduce the computation is confused, since the dense setting achives even faster inference speed. It looks that the dense representation obtains a better balance between performance and efficiency, while the results of dense setting are also used to calculate the improvements in abstract by the authors.
3. The comparision between previous works in Table 1 is unfair. Specially, the UniAD and VAD utilize different evaluation protocol as mentioned in PARA-Drive [2]. It looks this paper follows the approach in VAD without transforming UniAD results, which will obtain lower L2 error and collision rate. Therefore, the claimed performance improvements to UniAD are incorrect. Meanwhile, the collision rate of SparseDrive shows large gap between reported in its paper, which deserve to re-check.
4. The method is only evaluated in open-loop nuScenes dataset without close-loop simulation.

[2] Xinshuo Weng, Boris Ivanovic, Yan Wang, Yue Wang, and Marco Pavone. Para-drive: Parallelized architecture for real-time autonomous driving. In CVPR, 2024.

**Questions:**

1. What’s the relationship between agent features F_a, map features F_m and surrounding features F_o? Will map queries involved into the selection of interactive queries?
2. Will the path of other objects be used to calculate the geometric prior in query selection? There may be some objects drive far form the ego path, which is closet to the path but should be pick out. While some objects drive close to the ego path when lane changing, which are far from the path but should pay more attention to.
3. In Figure 4, why the ego query is updated by weighted object queries? Does that also mean the map queries will not affect the ego query feature?
4. The introduction of DiFSD(Dense) confuses me about the contribution in sparsity. As you claimed that dense scene representation is computationally intensive, why the dense-based setting is even faster than your fully sparse design (14.8 FPS vs 10.7 FPS)? Meanwhile, I thinke the latency comparison of w/ and w/o interactive query selection will help to illustrate the efficiency of the module.
5. The visualization of CIPV/CIPS in different scenes iteratively will help reader to understand the process of iterative motion planner. Will you provide some qualitative results?

---

### Note · Authors · 2024-11-14

I have read and agree with the venue's withdrawal policy on behalf of myself and my co-authors.